# Parallel Successive Convex Approximation for Nonsmooth Nonconvex Optimization

**Meisam Razaviyayn**[*]
meisamr@stanford.edu

**Mingyi Hong**[†]
mingyi@iastate.edu

**Zhi-Quan Luo**[‡]
luozq@umn.edu

**Jong-Shi Pang**[§]
jongship@usc.edu

## Abstract

Consider the problem of minimizing the sum of a smooth (possibly non-convex) and a convex (possibly nonsmooth) function involving a large number of variables. A popular approach to solve this problem is the block coordinate descent (BCD) method whereby at each iteration only one variable block is updated while the remaining variables are held fixed. With the recent advances in the developments of the multi-core parallel processing technology, it is desirable to parallelize the BCD method by allowing multiple blocks to be updated simultaneously at each iteration of the algorithm. In this work, we propose an inexact parallel BCD approach where at each iteration, a subset of the variables is updated in parallel by minimizing convex approximations of the original objective function. We investigate the convergence of this parallel BCD method for both randomized and cyclic variable selection rules. We analyze the asymptotic and non-asymptotic convergence behavior of the algorithm for both convex and non-convex objective functions. The numerical experiments suggest that for a special case of Lasso minimization problem, the cyclic block selection rule can outperform the randomized rule.

## 1 Introduction

Consider the following optimization problem

$$\min_x \quad h(x) \triangleq f(x_1, \ldots, x_n) + \sum_{i=1}^{n} g_i(x_i) \qquad \text{s.t. } x_i \in \mathcal{X}_i, \ i = 1, 2, \ldots, n, \tag{1}$$

where $\mathcal{X}_i \subseteq \mathbb{R}^{m_i}$ is a closed convex set; the function $f : \prod_{i=1}^{n} \mathcal{X}_i \to \mathbb{R}$ is a smooth function (possibly non-convex); and $g(x) \triangleq \sum_{i=1}^{n} g_i(x_i)$ is a separable convex function (possibly nonsmooth). The above optimization problem appears in various fields such as machine learning, signal processing, wireless communication, image processing, social networks, and bioinformatics, to name just a few. These optimization problems are typically of huge size and should be solved expeditiously.

A popular approach for solving the above multi-block optimization problem is the block coordinate descent (BCD) approach, where at each iteration of BCD, only one of the block variables is updated, while the remaining blocks are held fixed. Since only one block is updated at each iteration, the per-iteration storage and computational demand of the algorithm is low, which is desirable in huge-size problems. Furthermore, as observed in [1–3], these methods perform particulary well in practice.

---

[*]Electrical Engineering Department, Stanford University

[†]Industrial and Manufacturing Systems Engineering, Iowa State University

[‡]Department of Electrical and Computer Engineering, University of Minnesota

[§]Department of Industrial and Systems Engineering, University of Southern California

The availability of high performance multi-core computing platforms makes it increasingly desirable to develop parallel optimization methods. One category of such parallelizable methods is the (proximal) gradient methods. These methods are parallelizable in nature [4–8]; however, they are equivalent to successive minimization of a quadratic approximation of the objective function which may not be tight; and hence suffer from low convergence speed in some practical applications [9].

To take advantage of the BCD method and parallel multi-core technology, different parallel BCD algorithms have been recently proposed in the literature. In particular, the references [10–12] propose parallel coordinate descent minimization methods for $\ell_1$-regularized convex optimization problems. Using the greedy (Gauss-Southwell) update rule, the recent works [9,13] propose parallel BCD type methods for general composite optimization problems. In contrast, references [2,14–20] suggest randomized block selection rule, which is more amenable to big data optimization problems, in order to parallelize the BCD method.

Motivated by [1,9,15,21], we propose a parallel inexact BCD method where at each iteration of the algorithm, a subset of the blocks is updated by minimizing locally tight approximations of the objective function. Asymptotic and non-asymptotic convergence analysis of the algorithm is presented in both convex and non-convex cases for different variable block selection rules. The proposed parallel algorithm is synchronous, which is different than the existing lock-free methods in [22,23].

The contributions of this work are as follows:

- A parallel block coordinate descent method is proposed for non-convex nonsmooth problems. To the best of our knowledge, reference [9] is the only paper in the literature that focuses on parallelizing BCD for non-convex nonsmooth problems. This reference utilizes greedy block selection rule which requires search among all blocks as well as communication among processing nodes in order to find the best blocks to update. This requirement can be demanding in practical scenarios where the communication among nodes are costly or when the number of blocks is huge. In fact, this high computational cost motivated the authors of [9] to develop further inexact update strategies to efficiently alleviating the high computational cost of the greedy block selection rule.

- The proposed parallel BCD algorithm allows both cyclic and randomized block variable selection rules. The deterministic (*cyclic*) update rule is different than the existing parallel randomized or greedy BCD methods in the literature; see, e.g., [2,9,13–20]. Based on our numerical experiments, this update rule is beneficial in solving the Lasso problem.

- The proposed method not only works with the constant step-size selection rule, but also with the diminishing step-sizes which is desirable when the Lipschitz constant of the objective function is not known.

- Unlike many existing algorithms in the literature, e.g. [13–15], our parallel BCD algorithm utilizes the general approximation of the original function which includes the linear/proximal approximation of the objective as a special case. The use of general approximation instead of the linear/proximal approximation offers more flexibility and results in efficient algorithms for particular practical problems; see [21,24] for specific examples.

- We present an iteration complexity analysis of the algorithm for both convex and non-convex scenarios. Unlike the existing non-convex parallel methods in the literature such as [9] which only guarantee the asymptotic behavior of the algorithm, we provide non-asymptotic guarantees on the convergence of the algorithm as well.

## 2   Parallel Successive Convex Approximation

As stated in the introduction section, a popular approach for solving (1) is the BCD method where at iteration $r+1$ of the algorithm, the block variable $x_i$ is updated by solving the following subproblem

$$x_i^{r+1} = \arg \min_{x_i \in \mathcal{X}_i} \quad h(x_1^r, \ldots, x_{i-1}^r, x_i, x_{i+1}^r, \ldots, x_n^r). \tag{2}$$

In many practical problems, the update rule (2) is not in closed form and hence not computationally cheap. One popular approach is to replace the function $h(\cdot)$ with a well-chosen local convex

approximation $\widetilde{h}_i(x_i, x^r)$ in (2). That is, at iteration $r + 1$, the block variable $x_i$ is updated by

$$x_i^{r+1} = \arg\min_{x_i \in \mathcal{X}_i} \quad \widetilde{h}_i(x_i, x^r), \tag{3}$$

where $\widetilde{h}_i(x_i, x^r)$ is a convex (possibly upper-bound) approximation of the function $h(\cdot)$ with respect to the $i$-th block around the current iteration $x^r$. This approach, also known as *block successive convex approximation* or *block successive upper-bound minimization* [21], has been widely used in different applications; see [21, 24] for more details and different useful approximation functions. In this work, we assume that the approximation function $\widetilde{h}_i(\cdot, \cdot)$ is of the following form:

$$\widetilde{h}_i(x_i, y) = \widetilde{f}_i(x_i, y) + g_i(x_i). \tag{4}$$

Here $\widetilde{f}_i(\cdot, y)$ is an approximation of the function $f(\cdot)$ around the point $y$ with respect to the $i$-th block. We further assume that $\widetilde{f}_i(x_i, y) : \mathcal{X}_i \times \mathcal{X} \to \mathbb{R}$ satisfies the following assumptions:

- $\widetilde{f}_i(\cdot, y)$ is continuously differentiable and uniformly strongly convex with parameter $\tau$, i.e., $\widetilde{f}_i(x_i, y) \geq \widetilde{f}_i(x_i', y) + \langle \nabla_{x_i} \widetilde{f}_i(x_i', y), x_i - x_i' \rangle + \frac{\tau}{2}\|x_i - x_i'\|^2, \ \forall x_i, x_i' \in \mathcal{X}_i, \ \forall y \in \mathcal{X}$

- *Gradient consistency assumption*: $\nabla_{x_i} \widetilde{f}_i(x_i, x) = \nabla_{x_i} f(x), \ \forall x \in \mathcal{X}$

- $\nabla_{x_i} \widetilde{f}_i(x_i, \cdot)$ is Lipschitz continuous on $\mathcal{X}$ for all $x_i \in \mathcal{X}_i$ with constant $\widetilde{L}$, i.e., $\|\nabla_{x_i} \widetilde{f}_i(x_i, y) - \nabla_{x_i} \widetilde{f}_i(x_i, z)\| \leq \widetilde{L}\|y - z\|, \quad \forall y, z \in \mathcal{X}, \ \forall x_i \in \mathcal{X}_i, \ \forall i.$

For instance, the following traditional proximal/quadratic approximations of $f(\cdot)$ satisfy the above assumptions when the feasible set is compact and $f(\cdot)$ is twice continuously differentiable:

- $\widetilde{f}(x_i, y) = \langle \nabla_{y_i} f(y), x_i - y_i \rangle + \frac{\alpha}{2}\|x_i - y_i\|^2.$
- $\widetilde{f}(x_i, y) = f(x_i, y_{-i}) + \frac{\alpha}{2}\|x_i - y_i\|^2$, for $\alpha$ large enough.

For other practical useful approximations of $f(\cdot)$ and the stochastic/incremental counterparts, see [21, 25, 26].

With the recent advances in the development of parallel processing machines, it is desirable to take the advantage of multi-core machines by updating multiple blocks simultaneously in (3). Unfortunately, naively updating multiple blocks simultaneously using the approach (3) does not result in a convergent algorithm. Hence, we suggest to modify the update rule by using a well-chosen step-size. More precisely, we propose Algorithm 1 for solving the optimization problem (1).

---

**Algorithm 1** Parallel Successive Convex Approximation (PSCA) Algorithm

---

find a feasible point $x^0 \in \mathcal{X}$ and set $r = 0$
**for** $r = 0, 1, 2, \dots$ **do**
    choose a subset $S^r \subseteq \{1, \dots, n\}$
    calculate $\widehat{x}_i^r = \arg\min_{x_i \in \mathcal{X}_i} \widetilde{h}_i(x_i, x^r), \ \forall i \in S^r$
    set $x_i^{r+1} = x_i^r + \gamma^r(\widehat{x}_i^r - x_i^r), \ \forall i \in S^r$, and set $x_i^{r+1} = x_i^r, \ \forall i \notin S^r$
**end for**

---

The procedure of selecting the subset $S^r$ is intentionally left unspecified in Algorithm 1. This selection could be based on different rules. Reference [9] suggests the greedy variable selection rule where at each iteration of the algorithm in [9], the best response of all the variables are calculated and at the end, only the block variables with the largest amount of improvement are updated. A drawback of this approach is the overhead caused by the calculation of all of the best responses at each iteration; this overhead is especially computationally demanding when the size of the problem is huge. In contrast to [9], we suggest the following *randomized* or *cyclic* variable selection rules:

- **Cyclic**: Given the partition $\{\mathcal{T}_0, \dots, \mathcal{T}_{m-1}\}$ of the set $\{1, 2, \dots, n\}$ with $\mathcal{T}_i \bigcap \mathcal{T}_j = \emptyset, \ \forall i \neq j$ and $\bigcup_{\ell=0}^{m-1} \mathcal{T}_\ell = \{1, 2, \dots, n\}$, we say the choice of the variable selection is *cyclic* if

$$S^{mr+\ell} = \mathcal{T}_\ell, \quad \forall \ell = 0, 1, \dots, m - 1 \text{ and } \forall r,$$

- **Randomized:** The variable selection rule is called *randomized* if at each iteration the variables are chosen randomly from the previous iterations so that

$$Pr(j \in S^r \mid x^r, x^{r-1}, \ldots, x^0) = p_j^r \geq p_{\min} > 0, \quad \forall j = 1, 2, \ldots, n, \ \forall r.$$

## 3 Convergence Analysis: Asymptotic Behavior

We first make a standard assumption that $\nabla f(\cdot)$ is Lipschitz continuous with constant $L_{\nabla f}$, i.e.,

$$\|\nabla f(x) - \nabla f(y)\| \leq L_{\nabla f}\|x - y\|,$$

and assume that $-\infty < \inf_{x \in \mathcal{X}} h(x)$. Let us also define $\bar{x}$ to be a *stationary point* of (1) if $\exists d \in \partial g(\bar{x})$ such that $\langle \nabla f(\bar{x}) + d, x - \bar{x} \rangle \geq 0$, $\forall x \in \mathcal{X}$, i.e., the first order optimality condition is satisfied at the point $\bar{x}$. The following lemma will help us to study the asymptotic convergence of the PSCA algorithm.

**Lemma 1** *[9, Lemma 2] Define the mapping $\widehat{x}(\cdot) : \mathcal{X} \mapsto \mathcal{X}$ as $\widehat{x}(y) = (\widehat{x}_i(y))_{i=1}^n$ with $\widehat{x}_i(y) = \arg\min_{x_i \in \mathcal{X}_i} \widetilde{h}_i(x_i, y)$. Then the mapping $\widehat{x}(\cdot)$ is Lipschitz continuous with constant $\widehat{L} = \frac{\sqrt{n}\widetilde{L}}{\tau}$, i.e.,*

$$\|\widehat{x}(y) - \widehat{x}(z)\| \leq \widehat{L}\|y - z\|, \ \forall y, z \in \mathcal{X}.$$

Having derived the above result, we are now ready to state our first result which studies the limiting behavior of the PSCA algorithm. This result is based on the sufficient decrease of the objective function which has been also exploited in [9] for greedy variable selection rule.

**Theorem 1** *Assume $\gamma^r \in (0, 1]$, $\sum_{r=1}^{\infty} \gamma^r = +\infty$, and that $\limsup_{r \to \infty} \gamma^r < \bar{\gamma} \triangleq \min\{\frac{\tau}{L_{\nabla f}}, \frac{\tau}{\tau + \widetilde{L}\sqrt{n}}\}$. Suppose either cyclic or randomized block selection rule is employed. For cyclic update rule, assume further that $\{\gamma^r\}_{r=1}^{\infty}$ is a monotonically decreasing sequence. Then every limit point of the iterates is a stationary point of (1) – deterministically for cyclic update rule and almost surely for randomized block selection rule.*

**Proof** Using the standard sufficient decrease argument (see the supplementary materials), one can show that

$$h(x^{r+1}) \leq h(x^r) + \frac{\gamma^r(-\tau + \gamma^r L_{\nabla f})}{2}\|\widehat{x}^r - x^r\|_{S^r}^2. \tag{5}$$

Since $\limsup_{r \to \infty} \gamma^r < \bar{\gamma}$, for sufficiently large $r$, there exists $\beta > 0$ such that

$$h(x^{r+1}) \leq h(x^r) - \beta\gamma^r\|\widehat{x}^r - x^r\|_{S^r}^2. \tag{6}$$

Taking the conditional expectation from both sides implies

$$\mathbb{E}[h(x^{r+1}) \mid x^r] \leq h(x^r) - \beta\gamma^r \mathbb{E}\left[\sum_{i=1}^n R_i^r\|\widehat{x}_i^r - x_i^r\|^2 \mid x^r\right], \tag{7}$$

where $R_i^r$ is a Bernoulli random variable which is one if $i \in S^r$ and it is zero otherwise. Clearly, $\mathbb{E}[R_i^r \mid x^r] = p_i^r$ and therefore,

$$\mathbb{E}[h(x^{r+1}) \mid x^r] \leq h(x^r) - \beta\gamma^r p_{\min}\|\widehat{x}^r - x^r\|^2, \quad \forall r. \tag{8}$$

Thus $\{h(x^r)\}$ is a supermartingale with respect to the natural history; and by the supermartingale convergence theorem [27, Proposition 4.2], $h(x^r)$ converges and we have

$$\sum_{r=1}^{\infty} \gamma^r\|\widehat{x}^r - x^r\|^2 < \infty, \quad \text{almost surely.} \tag{9}$$

Let us now restrict our analysis to the set of probability one for which $h(x^r)$ converges and $\sum_{r=1}^{\infty} \gamma^r\|\widehat{x}^r - x^r\|^2 < \infty$. Fix a realization in that set. The equation (9) simply implies that, for the fixed realization, $\liminf_{r \to \infty} \|\widehat{x}^r - x^r\| = 0$, since $\sum_r \gamma^r = \infty$. Next we strengthen this result by proving that $\lim_{r \to \infty} \|\widehat{x}^r - x^r\| = 0$. Suppose the contrary that there exists $\delta > 0$ such

that $\Delta^r \triangleq \|\widehat{x}^r - x^r\| \geq 2\delta$ infinitely often. Since $\liminf_{r \to \infty} \Delta^r = 0$, there exists a subset of indices $\mathcal{K}$ and $\{i_r\}$ such that for any $r \in \mathcal{K}$,

$$\Delta^r < \delta, \quad 2\delta < \Delta^{i_r}, \quad \text{and} \quad \delta \leq \Delta^j \leq 2\delta, \; \forall j = r+1, \ldots, i_r - 1. \tag{10}$$

Clearly,

$$\delta - \Delta^r \overset{(i)}{\leq} \Delta^{r+1} - \Delta^r = \|\widehat{x}^{r+1} - x^{r+1}\| - \|\widehat{x}^r - x^r\| \overset{(ii)}{\leq} \|\widehat{x}^{r+1} - \widehat{x}^r\| + \|x^{r+1} - x^r\|$$

$$\overset{(iii)}{\leq} (1 + \widehat{L})\|x^{r+1} - x^r\| \overset{(iv)}{=} (1 + \widehat{L})\gamma^r\|\widehat{x}^r - x^r\| \leq (1 + \widehat{L})\gamma^r\delta, \tag{11}$$

where (i) and (ii) are due to (10) and the triangle inequality, respectively. The inequality (iii) is the result of Lemma 1; and (iv) is followed from the algorithm iteration update rule. Since $\limsup_{r \to \infty} \gamma^r < \frac{1}{1+\widehat{L}}$, the above inequality implies that there exists an $\alpha > 0$ such that

$$\Delta^r > \alpha, \tag{12}$$

for all $r$ large enough. Furthermore, since the chosen realization satisfies (9), we have that $\lim_{r \to \infty} \sum_{t=r}^{i_r - 1} \gamma^t (\Delta^t)^2 = 0$; which combined with (10) and (12), implies

$$\lim_{r \to \infty} \sum_{t=r}^{i_r - 1} \gamma^t = 0. \tag{13}$$

On the other hand, using the similar reasoning as in above, one can write

$$\delta < \Delta^{i_r} - \Delta^r = \|\widehat{x}^{i_r} - x^{i_r}\| - \|\widehat{x}^r - x^r\| \leq \|\widehat{x}^{i_r} - \widehat{x}^r\| + \|x^{i_r} - x^r\|$$

$$\leq (1 + \widehat{L}) \sum_{t=r}^{i_r - 1} \gamma^t \|\widehat{x}^t - x^t\| \leq 2\delta(1 + \widehat{L}) \sum_{t=r}^{i_r - 1} \gamma^t,$$

and hence $\liminf_{r \to \infty} \sum_{t=r}^{i_r - 1} \gamma^t > 0$, which contradicts (13). Therefore the contrary assumption does not hold and we must have $\lim_{r \to \infty} \|\widehat{x}^r - x^r\| = 0$, almost surely. Now consider a limit point $\bar{x}$ with the subsequence $\{x^{r_j}\}_{j=1}^{\infty}$ converging to $\bar{x}$. Using the definition of $\widehat{x}^{r_j}$, we have $\lim_{j \to \infty} \widetilde{h}_i(\widehat{x}_i^{r_j}, x^{r_j}) \leq \widetilde{h}_i(x_i, x^{r_j})$, $\forall x_i \in \mathcal{X}_i$, $\forall i$. Therefore, by letting $j \to \infty$ and using the fact that $\lim_{r \to \infty} \|\widehat{x}^r - x^r\| = 0$, almost surely, we obtain $\widetilde{h}_i(\bar{x}_i, \bar{x}) \leq \widetilde{h}_i(x_i, \bar{x})$, $\forall x_i \in \mathcal{X}_i$, $\forall i$, almost surely; which in turn, using the gradient consistency assumption, implies

$$\langle \nabla f(\bar{x}) + d, x - \bar{x} \rangle \geq 0, \; \forall x \in \mathcal{X}, \; \text{almost surely,}$$

for some $d \in \partial g(\bar{x})$, which completes the proof for the randomized block selection rule.

Now consider the cyclic update rule with a limit point $\bar{x}$. Due to the sufficient decrease bound (6), we have $\lim_{r \to \infty} h(x^r) = h(\bar{x})$. Furthermore, by taking the summation over (6), we obtain $\sum_{r=1}^{\infty} \gamma^r \|\widehat{x}^r - x^r\|_{S^r}^2 < \infty$. Consider a fixed block $i$ and define $\{r_k\}_{k=1}^{\infty}$ to be the subsequence of iterations that block $i$ is updated in. Clearly, $\sum_{k=1}^{\infty} \gamma^{r_k} \|\widehat{x}_i^{r_k} - x_i^{r_k}\|^2 < \infty$ and $\sum_{k=1}^{\infty} \gamma^{r_k} = \infty$, since $\{\gamma^r\}$ is monotonically decreasing. Therefore, $\liminf_{k \to \infty} \|\widehat{x}_i^{r_k} - x_i^{r_k}\| = 0$. Repeating the above argument with some slight modifications, which are omitted due to lack of space, we can show that $\lim_{k \to \infty} \|\widehat{x}_i^{r_k} - x_i^{r_k}\| = 0$ implying that the limit point $\bar{x}$ is a stationary point of (1). ∎

**Remark 1** *Theorem 1 covers both diminishing and constant step-size selection rule; or the combination of the two, i.e., decreasing the step-size until it is less than the constant $\bar{\gamma}$. It is also worth noting that the diminishing step-size rule is especially useful when the knowledge of the problem's constants $L, \widetilde{L}$, and $\tau$ is not available.*

## 4 Convergence Analysis: Iteration Complexity

In this section, we present iteration complexity analysis of the algorithm for both convex and non-convex cases.

## 4.1 Convex Case

When the function $f(\cdot)$ is convex, the overall objective function will become convex; and as a result of Theorem 1, if a limit point exists, it is a global minimizer of (1). In this scenario, it is desirable to derive the iteration complexity bounds of the algorithm. Note that our algorithm employs linear combination of the two consecutive points at each iteration and hence it is different than the existing algorithms in [2, 14–20]. Therefore, not only in the cyclic case, but also in the randomized scenario, the iteration complexity analysis of PSCA is different than the existing results and should be investigated. Let us make the following assumptions for our iteration complexity analysis:

- The step-size is constant with $\gamma^r = \gamma < \frac{\tau}{L_{\nabla f}}$, $\forall r$.

- The level set $\{x \mid h(x) \leq h(x^0)\}$ is compact and the next two assumptions hold in this set.

- The nonsmooth function $g(\cdot)$ is Lipschitz continuous, i.e., $|g(x) - g(y)| \leq L_g\|x - y\|$, $\forall x, y \in \mathcal{X}$. This assumption is satisfied in many practical problems such as (group) Lasso.

- The gradient of the approximation function $\widetilde{f}_i(\cdot, y)$ is uniformly Lipschitz with constant $L_i$, i.e., $\|\nabla_{x_i}\widetilde{f}_i(x_i, y) - \nabla_{x_i'}\widetilde{f}_i(x_i', y)\| \leq L_i\|x_i - x_i'\|$, $\forall x_i, x_i' \in \mathcal{X}_i$.

**Lemma 2 (Sufficient Descent)** *There exists $\widehat{\beta}, \widetilde{\beta} > 0$, such that for all $r \geq 1$, we have*

- *For randomized rule:* $\mathbb{E}[h(x^{r+1}) \mid x^r] \leq h(x^r) - \widehat{\beta}\|\widehat{x}^r - x^r\|^2$.

- *For cyclic rule:* $h(x^{m(r+1)}) \leq h(x^{mr}) - \widetilde{\beta}\|x^{m(r+1)} - x^{mr}\|^2$.

**Proof** The above result is an immediate consequence of (6) with $\widehat{\beta} \triangleq \beta\gamma p_{\min}$ and $\widetilde{\beta} \triangleq \frac{\beta}{\gamma}$. ∎

Due to the bounded level set assumption, there must exist constants $\widehat{Q}, Q, R > 0$ such that

$$\|\nabla f(x^r)\| \leq Q, \qquad \|\nabla_{x_i}\widetilde{f}_i(\widehat{x}^r, x^r)\| \leq \widehat{Q}, \qquad \|x^r - x^*\| \leq R, \qquad (14)$$

for all $x^r$. Next we use the constants $Q, \widehat{Q}$ and $R$ to bound the cost-to-go in the algorithm.

**Lemma 3 (Cost-to-go Estimate)** *For all $r \geq 1$, we have*

- *For randomized rule:* $\left(\mathbb{E}[h(x^{r+1}) \mid x^r] - h(x^*)\right)^2 \leq 2\left((Q + L_g)^2 + nL^2R^2\right)\|\widehat{x}^r - x^r\|^2$

- *For cyclic rule:* $\left(h(x^{m(r+1)}) - h(x^*)\right)^2 \leq 3n\frac{\theta(1-\gamma)^2}{\gamma^2}\|x^{m(r+1)} - x^{mr}\|^2$

*for any optimal point $x^*$, where $L \triangleq \max_i\{L_i\}$ and $\theta \triangleq L_g^2 + \widehat{Q}^2 + 2nR^2\tilde{L}^2\frac{\gamma^2}{(1-\gamma)^2} + 2R^2L^2$.*

**Proof** Please see the supplementary materials for the proof.

Lemma 2 and Lemma 3 lead to the iteration complexity bound in the following theorem. The proof steps of this result are similar to the ones in [28] and therefore omitted here for space reasons.

**Theorem 2** *Define $\sigma \triangleq \frac{(\gamma L_{\nabla f} - \tau)\gamma p_{\min}}{4((Q+L_g)^2 + nL^2R^2)}$ and $\widetilde{\sigma} \triangleq \frac{(\gamma L_{\nabla f} - \tau)\gamma}{6n\theta(1-\gamma)^2}$. Then*

- *For randomized update rule:* $\mathbb{E}[h(x^r)] - h(x^*) \leq \frac{\max\{4\sigma-2, h(x^0)-h(x^*), 2\}}{\sigma}\frac{1}{r}$.

- *For cyclic update rule:* $h(x^{mr}) - h(x^*) \leq \frac{\max\{4\widetilde{\sigma}-2, h(x^0)-h(x^*), 2\}}{\widetilde{\sigma}}\frac{1}{r}$.

## 4.2 Non-convex Case

In this subsection we study the iteration complexity of the proposed randomized algorithm for the general nonconvex function $f(\cdot)$ assuming constant step-size selection rule. This analysis is only for the randomized block selection rule. Since in the nonconvex scenario, the iterates may not converge to the global optimum point, the closeness to the optimal solution cannot be considered for the iteration complexity analysis. Instead, inspired by [29] where the size of the gradient of the objective function is used as a measure of optimality, we consider the size of the objective proximal gradient as a measure of optimality. More precisely, we define

$$\widetilde{\nabla} h(x) = x - \arg\min_{y \in \mathcal{X}} \left\{ \langle \nabla f(x), y - x \rangle + g(y) + \frac{1}{2} \|y - x\|^2 \right\}.$$

Clearly, $\widetilde{\nabla} h(x) = 0$ when $x$ is a stationary point. Moreover, $\widetilde{\nabla} h(\cdot)$ coincides with the gradient of the objective if $g \equiv 0$ and $\mathcal{X} = \mathbb{R}^n$. The following theorem, which studies the decrease rate of $\|\widetilde{\nabla} h(x)\|$, could be viewed as an iteration complexity analysis of the randomized PSCA.

**Theorem 3** *Consider randomized block selection rule. Define $T_\epsilon$ to be the first time that* $\mathbb{E}[\|\widetilde{\nabla} h(x^r)\|^2] \leq \epsilon$. *Then $T_\epsilon \leq \kappa/\epsilon$ where $\kappa \triangleq \frac{2(L^2+2L+2)(h(x^0)-h^*)}{\widehat{\beta}}$ and $h^* = \min_{x \in \mathcal{X}} h(x)$.*

**Proof** To simplify the presentation of the proof, let us define $\widetilde{y}_i^r \triangleq \arg\min_{y_i \in \mathcal{X}_i} \langle \nabla_{x_i} f(x^r), y_i - x_i^r \rangle + g_i(y_i) + \frac{1}{2}\|y_i - x_i^r\|^2$. Clearly, $\widetilde{\nabla} h(x^r) = (x_i^r - \widetilde{y}_i^r)_{i=1}^n$. The first order optimality condition of the above optimization problem implies

$$\langle \nabla_{x_i} f(x^r) + \widetilde{y}_i^r - x_i^r, x_i - \widetilde{y}_i^r \rangle + g_i(x_i) - g_i(\widetilde{y}_i^r) \geq 0, \quad \forall x_i \in \mathcal{X}_i. \tag{15}$$

Furthermore, based on the definition of $\widehat{x}_i^r$, we have

$$\langle \nabla_{x_i} \widetilde{f}_i(\widehat{x}_i^r, x^r), x_i - \widehat{x}_i^r \rangle + g_i(x_i) - g_i(\widehat{x}_i^r) \geq 0, \quad \forall x_i \in \mathcal{X}_i. \tag{16}$$

Plugging in the points $\widehat{x}_i^r$ and $\widetilde{y}_i^r$ in (15) and (16); and summing up the two equations will yield to

$$\langle \nabla_{x_i} \widetilde{f}_i(\widehat{x}_i^r, x^r) - \nabla_{x_i} f(x^r) + x_i^r - \widetilde{y}_i^r, \widetilde{y}_i^r - \widehat{x}_i^r \rangle \geq 0.$$

Using the gradient consistency assumption, we can write

$$\langle \nabla_{x_i} \widetilde{f}_i(\widehat{x}_i^r, x^r) - \nabla_{x_i} \widetilde{f}_i(x_i^r, x^r) + x_i^r - \widehat{x}_i^r + \widehat{x}_i^r - \widetilde{y}_i^r, \widetilde{y}_i^r - \widehat{x}_i^r \rangle \geq 0,$$

or equivalently, $\langle \nabla_{x_i} \widetilde{f}_i(\widehat{x}_i^r, x^r) - \nabla_{x_i} \widetilde{f}_i(x_i^r, x^r) + x_i^r - \widehat{x}_i^r, \widetilde{y}_i^r - \widehat{x}_i^r \rangle \geq \|\widehat{x}_i^r - \widetilde{y}_i^r\|^2$. Applying Cauchy-Schwarz and the triangle inequality will yield to

$$\left( \|\nabla_{x_i} \widetilde{f}_i(\widehat{x}_i^r, x^r) - \nabla_{x_i} \widetilde{f}_i(x_i^r, x^r)\| + \|x_i^r - \widehat{x}_i^r\| \right) \|\widetilde{y}_i^r - \widehat{x}_i^r\| \geq \|\widehat{x}_i^r - \widetilde{y}_i^r\|^2.$$

Since the function $\widetilde{f}_i(\cdot, x)$ is Lipschitz, we must have

$$\|\widehat{x}_i^r - \widetilde{y}_i^r\| \leq (1 + L_i)\|x_i^r - \widehat{x}_i^r\| \tag{17}$$

Using the inequality (17), the norm of the proximal gradient of the objective can be bounded by

$$\|\widetilde{\nabla} h(x^r)\|^2 = \sum_{i=1}^n \|x_i^r - \widetilde{y}_i^r\|^2 \leq 2 \sum_{i=1}^n \left( \|x_i^r - \widehat{x}_i^r\|^2 + \|\widehat{x}_i^r - \widetilde{y}_i^r\|^2 \right)$$

$$\leq 2 \sum_{i=1}^n \left( \|x_i^r - \widehat{x}_i^r\|^2 + (1 + L_i)^2 \|x_i^r - \widehat{x}_i^r\|^2 \right) \leq 2(2 + 2L + L^2)\|\widehat{x}^r - x^r\|^2.$$

Combining the above inequality with the sufficient decrease bound in (7), one can write

$$\sum_{r=0}^T \mathbb{E}\left[ \|\widetilde{\nabla} h(x^r)\|^2 \right] \leq \sum_{r=1}^T 2(2 + 2L + L^2) \mathbb{E}\left[ \|\widehat{x}^r - x^r\|^2 \right]$$

$$\leq \sum_{r=0}^T \frac{2(2 + 2L + L^2)}{\widehat{\beta}} \mathbb{E}\left[ h(x^r) - h(x^{r+1}) \right] \leq \frac{2(2 + 2L + L^2)}{\widehat{\beta}} \mathbb{E}\left[ h(x^0) - h(x^{T+1}) \right]$$

$$\leq \frac{2(2 + 2L + L^2)}{\widehat{\beta}} \left[ h(x^0) - h^* \right] = \kappa,$$

which implies that $T_\epsilon \leq \frac{\kappa}{\epsilon}$. ∎

## 5  Numerical Experiments:

In this short section, we compare the numerical performance of the proposed algorithm with the classical serial BCD methods. The algorithms are evaluated over the following Lasso problem:

$$\min_{x} \quad \frac{1}{2}\|Ax - b\|_2^2 + \lambda\|x\|_1,$$

where the matrix $A$ is generated according to the Nesterov's approach [5]. Two problem instances are considered: $A \in \mathbb{R}^{2000 \times 10,000}$ with 1% sparsity level in $x^*$ and $A \in \mathbb{R}^{1000 \times 100,000}$ with 0.1% sparsity level in $x^*$. The approximation functions are chosen similar to the numerical experiments in [9], i.e., block size is set to one ($m_i = 1, \ \forall i$) and the approximation function

$$\widetilde{f}(x_i, y) = f(x_i, y_{-i}) + \frac{\alpha}{2}\|x_i - y_i\|^2$$

is considered, where $f(x) = \frac{1}{2}\|Ax - b\|^2$ is the smooth part of the objective function. We choose constant step-size $\gamma$ and proximal coefficient $\alpha$. In general, careful selection of the algorithm parameters results in better numerical convergence rate. The smaller values of step-size $\gamma$ will result in less zigzag behavior for the convergence path of the algorithm; however, too small step sizes will clearly slow down the convergence speed. Furthermore, in order to make the approximation function sufficiently strongly convex, we need to choose $\alpha$ large enough. However, choosing too large $\alpha$ values enforces the next iterates to stay close to the current iterate and results in slower convergence speed; see the supplementary materials for related examples.

Figure 1 and Figure 2 illustrate the behavior of cyclic and randomized parallel BCD method as compared with their serial counterparts. The serial methods "Cyclic BCD" and "Randomized BCD" are based on the update rule in (2) with the cyclic and randomized block selection rules, respectively. The variable $q$ shows the number of processors and on each processor we update 40 scalar variables in parallel. As can be seen in Figure 1 and Figure 2, parallelization of the BCD algorithm results in more efficient algorithm. However, the computational gain does not grow linearly with the number of processors. In fact, we can see that after some point, the increase in the number of processors lead to slower convergence. This fact is due to the communication overhead among the processing nodes which dominates the computation time; see the supplementary materials for more numerical experiments on this issue.

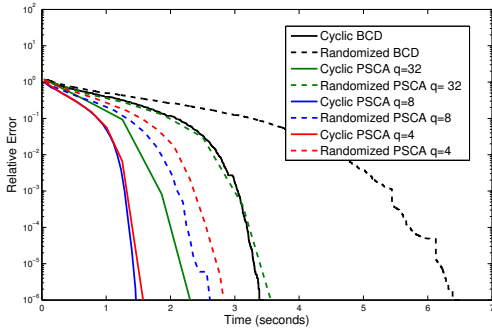

Figure 1: Lasso Problem: $A \in \mathbb{R}^{2,000 \times 10,000}$

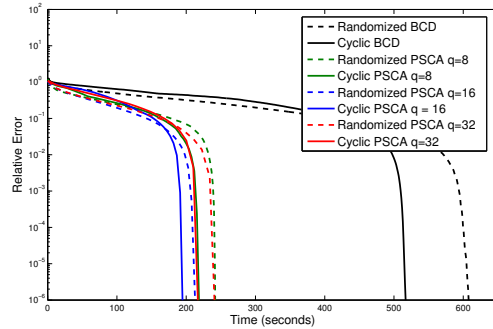

Figure 2: Lasso Problem: $A \in \mathbb{R}^{1,000 \times 100,000}$

**Acknowledgments:** The authors are grateful to the University of Minnesota Graduate School Doctoral Dissertation Fellowship and AFOSR, grant number FA9550-12-1-0340 for the support during this research.

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
