[Supplementary Material]

# Supplementary Material for Parallel Successive Convex Approximation for Nonsmooth Nonconvex Optimization

**Meisam Razaviyayn**[*]
meisamr@stanford.edu

**Mingyi Hong**[†]
mingyi@iastate.edu

**Zhi-Quan Luo** [‡]
luozq@umn.edu

**Jong-Shi Pang**[§]
jongship@usc.edu

## 1 Proofs

**Proof of Equation (5):**

Let us first show the result for the randomized block selection rule. We will do so by proving that $\lim_{r\to\infty} \|\widehat{x}^r - x^r\| = 0$, with probability one. To show this, we start by bounding the decrease in the objective value in the consecutive steps of the algorithm:

$$h(x^{r+1}) = f(x^{r+1}) + \sum_i g_i(x_i^{r+1}) = f(x^{r+1}) + \sum_{i\notin S^r} g_i(x_i^r) + \sum_{i\in S^r} g_i\left(x_i^r + \gamma^r(\widehat{x}_i^r - x_i^r)\right)$$

$$\leq f(x^{r+1}) + \sum_i g_i(x_i^r) + \gamma^r \sum_{i\in S^r} \left(g_i(\widehat{x}_i^r) - g_i(x_i^r)\right)$$

$$\leq f(x^r) + \gamma^r \langle \nabla f(x^r), \widehat{x}^r - x^r\rangle_{S^r} + \frac{(\gamma^r)^2 L_{\nabla F}}{2}\|\widehat{x}^r - x^r\|_{S^r}^2 + \sum_i g_i(x_i^r) + \gamma^r \sum_{i\in S^r}\left(g_i(\widehat{x}_i^r) - g_i(x_i^r)\right)$$

$$= h(x^r) + \frac{(\gamma^r)^2 L_{\nabla f}}{2}\|\widehat{x}^r - x^r\|_{S^r}^2 + \gamma^r\left(\langle\nabla f(x^r),\widehat{x}^r - x^r\rangle_{S^r} + \sum_{i\in S^r}\left(g_i(\widehat{x}_i^r) - g_i(x_i^r)\right)\right), \tag{1}$$

where the first inequality is due to convexity of $g(\cdot)$; the second inequality is due to the Lipschitz continuity of $\nabla f(\cdot)$; and we have also used the notation $\langle a,b\rangle_S \triangleq \sum_{i\in S}\langle a_i,b_i\rangle$ and $\|a\|_S^2 \triangleq \langle a,a\rangle_S$. In order to get a standard form sufficient decrease bound, we need to bound the last term in (1). Noticing that $\widetilde{h}_i$ is strongly convex, the definition of $\widehat{x}_i^r$ leads to

$$\widetilde{h}_i(x_i^r, x^r) \geq \widetilde{h}_i(\widehat{x}_i^r, x^r) + \frac{\tau}{2}\|\widehat{x}_i^r - x_i^r\|^2, \ \forall i \in S^r.$$

Substituting the definition of $\widetilde{h}_i$ and multiplying both sides by minus one give

$$-\widetilde{f}_i(x_i^r, x^r) - g_i(x_i^r) \leq -\widetilde{f}_i(\widehat{x}_i^r, x^r) - g_i(\widehat{x}_i^r) - \frac{\tau}{2}\|\widehat{x}_i^r - x_i^r\|^2.$$

Linearizing the smooth part, the gradient consistency assumption leads to

$$\langle\nabla_{x_i}f(x^r), \widehat{x}_i^r - x_i^r\rangle + g_i(\widehat{x}_i^r) - g_i(x_i^r) \leq -\frac{\tau}{2}\|\widehat{x}_i^r - x_i^r\|^2.$$

Summing up the above inequality over all $i \in S^r$, we obtain

$$\langle\nabla_x f(x^r), \widehat{x}^r - x^r\rangle_{S^r} + \sum_{i\in S^r}\left(g_i(\widehat{x}_i^r) - g_i(x_i^r)\right) \leq -\frac{\tau}{2}\|\widehat{x}^r - x^r\|_{S^r}^2, \tag{2}$$

[*]Electrical Engineering Department, Stanford University

[†]Industrial and Manufacturing Systems Engineering, Iowa State University

[‡]Department of Electrical and Computer Engineering, University of Minnesota

[§]Department of Industrial and Systems Engineering, University of Southern California

where $\widehat{x}^r \triangleq (\widehat{x}_i^r)_{i=1}^n$. Combining (1) and (2) leads to

$$h(x^{r+1}) \le h(x^r) + \frac{\gamma^r(-\tau + \gamma^r L_{\nabla f})}{2}\|\widehat{x}^r - x^r\|_{S^r}^2.$$

**Proof of Lemma 3:**

Let us first prove the cost-to-go bound for the randomized case. It can be observed that the conditional expected cost-to-go can be bounded by

$$\mathbb{E}\left[h(x^{r+1}) - h(x^*) \mid x^r\right] \overset{(i)}{\le} h(x^r) - h(x^*) = f(x^r) - f(x^*) + g(x^r) - g(x^*)$$

$$\overset{(ii)}{\le} \langle \nabla f(x^r), x^r - \widehat{x}^r\rangle + \langle \nabla f(x^r), \widehat{x}^r - x^*\rangle + L_g\|x^r - \widehat{x}^r\| + g(\widehat{x}^r) - g(x^*)$$

$$\overset{(iii)}{\le} (L_g + Q)\|\widehat{x}^r - x^r\| + \sum_{i=1}^n \langle \nabla_{x_i} f(x^r) - \nabla_{x_i}\widetilde{f}_i(\widehat{x}_i, x^r) + \nabla_{x_i}\widetilde{f}_i(\widehat{x}_i, x^r), \widehat{x}_i^r - x_i^*\rangle + g(\widehat{x}^r) - g(x^*)$$

$$\le (L_g + Q)\|\widehat{x}^r - x^r\| + \sum_{i=1}^n \langle \nabla_{x_i} f(x^r) - \nabla_{x_i}\widetilde{f}_i(\widehat{x}_i, x^r), \widehat{x}_i^r - x_i^*\rangle \tag{3}$$

where (i) is due to the Sufficient Descent Lemma; the inequality (ii) is due to the convexity of $f(\cdot)$ and Lipschitz continuity of $g(\cdot)$; the third inequality is by the bounded level set assumption. Furthermore, the last inequality is obtained by exploiting the first order optimality condition of the point $\widehat{x}_i^r$, i.e., $\langle \nabla_{x_i}\widetilde{f}_i(\widehat{x}_i^r, x^r), \widehat{x}_i^r - x_i^*\rangle + g_i(\widehat{x}_i^r) - g_i(x_i^*) \le 0$. In addition to the above inequality, on can easily deduce

$$\left(\sum_{i=1}^n \langle \nabla_{x_i} f(x^r) - \nabla_{x_i}\widetilde{f}_i(\widehat{x}_i^r, x^r), \widehat{x}_i^r - x_i^*\rangle\right)^2 = \left(\sum_{i=1}^n \langle \nabla_{x_i}\widetilde{f}_i(x_i^r, x^r) - \nabla_{x_i}\widetilde{f}_i(\widehat{x}_i^r, x^r), \widehat{x}_i^r - x_i^*\rangle\right)^2$$

$$\le n\sum_{i=1}^n L_i^2\|x_i^r - \widehat{x}_i^r\|^2 \cdot \|\widehat{x}_i^r - x_i^*\|^2 \le nL^2R^2\|x^r - \widehat{x}^r\|^2. \tag{4}$$

Combining (3) and (4) will conclude the proof for the randomized case.

For the cyclic case, we can simply bound the cost-to-go estimate by

$$h(x^{m(r+1)}) - h(x^*) = f(x^{m(r+1)}) - f(x^*) + g(x^{m(r+1)}) - g(x^*)$$

$$\le \langle \nabla f(x^{m(r+1)}), x^{m(r+1)} - x^*\rangle + g(x^{m(r+1)}) - g(x^*) \tag{5}$$

$$= \sum_{\ell=0}^{m-1}\sum_{i\in\mathcal{T}_\ell} \langle \nabla_i f(x^{m(r+1)}), x_i^{m(r+1)} - x_i^*\rangle + g_i(x^{m(r+1)}) - g_i(x^*)$$

$$\le \sum_{\ell=0}^{m-1}\sum_{i\in\mathcal{T}_\ell} \langle \nabla_i f(x^{m(r+1)}) - \nabla_i\widetilde{f}_i(\widehat{x}_i^{mr+\ell}, x^{mr+\ell}), x_i^{m(r+1)} - x_i^*\rangle \tag{6}$$

$$+ \sum_{\ell=0}^{m-1}\sum_{i\in\mathcal{T}_\ell} \langle \nabla_i\widetilde{f}_i(\widehat{x}_i^{mr+\ell}, x^{mr+\ell}), x_i^{m(r+1)} - \widehat{x}_i^{mr+\ell}\rangle \tag{7}$$

$$+ \sum_{\ell=0}^{m-1}\sum_{i\in\mathcal{T}_\ell} \langle \nabla_i\widetilde{f}_i(\widehat{x}_i^{mr+\ell}, x^{mr+\ell}), \widehat{x}_i^{mr+\ell} - x_i^*\rangle + g_i(\widehat{x}_i^{mr+\ell}) - g_i(x_i^*) \tag{8}$$

$$+ \sum_{\ell=0}^{m-1}\sum_{i\in\mathcal{T}_\ell} g_i(x_i^{m(r+1)}) - g_i(\widehat{x}_i^{mr+\ell}), \tag{9}$$

where (5) is due to convexity of the function $f(\bullet)$. First notice that (8) is nonpositive due to the definition of $\widehat{x}_i^{mr+\ell}$ and the optimality condition for it. Now we bound the terms (6), (7), and (9) separately. Let us first start by bounding the terms in (6):

$$\left(\sum_{\ell=0}^{m-1}\sum_{i\in\mathcal{T}_\ell}\langle\nabla_i f(x^{m(r+1)})-\nabla_i\widetilde{f}_i(\widehat{x}_i^{mr+\ell},x^{mr+\ell}),x_i^{m(r+1)}-x_i^*\rangle\right)^2$$

$$\leq n\sum_{\ell=0}^{m-1}\sum_{i\in\mathcal{T}_\ell}R^2\|\nabla_i f(x^{m(r+1)})-\nabla_i\widetilde{f}_i(\widehat{x}_i^{mr+\ell},x^{mr+\ell})\|^2$$

$$= nR^2\sum_{\ell=0}^{m-1}\sum_{i\in\mathcal{T}_\ell}\|\nabla_i\widetilde{f}_i(x_i^{m(r+1)},x^{m(r+1)})-\nabla_i\widetilde{f}_i(\widehat{x}_i^{mr+\ell},x^{mr+\ell})\|^2$$

$$\leq 2nR^2\sum_{\ell=0}^{m-1}\sum_{i\in\mathcal{T}_\ell}\Big(\|\nabla_i\widetilde{f}_i(x_i^{m(r+1)},x^{m(r+1)})-\nabla_i\widetilde{f}_i(x_i^{m(r+1)},x^{mr+\ell})\|^2$$

$$+\|\nabla_i\widetilde{f}_i(x_i^{m(r+1)},x^{mr+\ell})-\nabla_i\widetilde{f}_i(\widehat{x}_i^{mr+\ell},x^{mr+\ell})\|^2\Big)$$

$$\leq 2nR^2\sum_{\ell=0}^{m-1}\sum_{i\in\mathcal{T}_\ell}\Big(\widetilde{L}^2\|x^{m(r+1)}-x^{mr+\ell}\|^2+L_i^2\|x_i^{m(r+1)}-\widehat{x}_i^{mr+\ell}\|^2\Big)$$

$$\leq 2nR^2\left(n\widetilde{L}^2\|x^{m(r+1)}-x^{mr}\|^2+L^2\frac{(1-\gamma)^2}{\gamma^2}\|x^{m(r+1)}-x^{mr}\|^2\right)$$

$$= 2nR^2\left(n\widetilde{L}^2+\frac{L^2(1-\gamma)^2}{\gamma^2}\right)\|x^{m(r+1)}-x^{mr}\|^2. \tag{10}$$

Now, we can bound (7) by

$$\left(\sum_{\ell=0}^{m-1}\sum_{i\in\mathcal{T}_\ell}\langle\nabla_i\widetilde{f}_i(\widehat{x}_i^{mr+\ell},x^{mr+\ell}),x_i^{m(r+1)}-\widehat{x}_i^{mr+\ell}\rangle\right)^2$$

$$\leq n\sum_{\ell=0}^{m-1}\sum_{i\in\mathcal{T}_\ell}\left(\langle\nabla_i\widetilde{f}_i(\widehat{x}^{mr+\ell},x^{mr+\ell}),x_i^{m(r+1)}-\widehat{x}_i^{mr+\ell}\rangle\right)^2$$

$$\leq n\sum_{\ell=0}^{m-1}\sum_{i\in\mathcal{T}_\ell}\widehat{Q}^2\|x_i^{m(r+1)}-\widehat{x}_i^{mr+\ell}\|^2$$

$$= n\widehat{Q}^2\sum_i\frac{(1-\gamma)^2}{\gamma^2}\|x_i^{m(r+1)}-x_i^{mr}\|^2$$

$$= n\widehat{Q}^2\frac{(1-\gamma)^2}{\gamma^2}\|x^{m(r+1)}-x^{mr}\|^2. \tag{11}$$

Finally, we can bound (9) by

$$\left(\sum_{\ell=0}^{m-1}\sum_{i\in\mathcal{T}_\ell}g_i(x_i^{m(r+1)})-g_i(\widehat{x}_i^{mr+\ell})\right)^2\leq n\sum_{\ell=0}^{m-1}\sum_{i\in\mathcal{T}_\ell}\left(g_i(x_i^{m(r+1)})-g_i(\widehat{x}_i^{mr+\ell})\right)^2$$

$$\leq n\sum_{\ell=0}^{m-1}\sum_{i\in\mathcal{T}_\ell}L_g^2\|x_i^{m(r+1)}-\widehat{x}^{mr+\ell}\|^2$$

$$= nL_g^2\sum_i\frac{(1-\gamma)^2}{\gamma^2}\|x_i^{m(r+1)}-x_i^{mr}\|^2$$

$$= nL_g^2\frac{(1-\gamma)^2}{\gamma^2}\|x^{m(r+1)}-x^{mr}\|^2. \tag{12}$$

Plugging (10), (11), (12) in (6), (7), (9) implies the desired cost-to-go bound.

# 2 Additional Numerical Experiments

In this section, we present two additional numerical experiments. Similar to the main body of the manuscript, we consider LASSO problem and the data is generated as explained before. Here we only consider the large size experiment, i.e., $A \in \mathbb{R}^{10,000 \times 100,000}$. In the first simulation, we see the effect of changing the parameters of the algorithm. As discussed in the paper, too large or too small step-size could result in slower convergence speed.

Figure 1: Performance of the algorithm with different choices of $\alpha$ and $\gamma$

As we saw in the main body of the manuscript, the overall convergence time of the algorithm does not always improve as the number of processors increases. This fact is due to the communication overhead among the processing nodes. In Figure 2, we only plot the computation time and ignore the communication time. As can be seen in this plot, the computation time spent on the nodes always decreases by utilizing more processors.

Figure 2: Computation time of the algorithm for different number of processors