[Reviews · NeurIPS 2014]

Submitted by Assigned_Reviewer_19

***** UPDATE ******
I thank the authors for their detailed response. I believe that the proposed modifications go into the right direction and will make the paper more typical and interesting for the NIPS conference.
*******************

The paper extends the block coordinate descent algorithm of [21] to a
composite and parallel setting. The convergence of the algorithm is
studied from a convex and non-convex perspectives, and the analysis
provides per-iteration complexities for randomized and cyclic update
rules.

On the plus side, the results are interesting, the method is flexible,
and the convergence analysis applies to a large variety of settings.
I also checked the convergence proofs and the analysis seems sound to me.
On the down side, the experimental validation is a bit weak and would
benefit from additional experiments on real data. This would require
gaining some space in the paper, which can be achieved by either
shortening the proof of Theorem 1 (see comments below), or relegating
some technical details to the appendix.

Misc comments:
- two pages of paper are devoted to the proof of Theorem 1, which is
neither the most interesting aspect of the paper, nor containing
particularly novel proof techniques. The proof could be shortened by
invoking a simple Lemma on converging sums (see Lemma A.5 in
http://arxiv.org/abs/1306.4650), applied to (10) and (12). Such
details could be also relegated to a supplementary material.
- even though the proof for the cyclic rule is lenghty (page 6), it
could be provided as supplementary material.
- regarding the ``impact score'', I would neither say that this work
will have a ``major impact'', but also would not say that it is
``incremental'' since it presents new results that are significant
enough.
Summary: The paper presents a parallel block coordinate descent
algorithm for composite optimization that is relevant for machine
learning. The algorithm enjoys nice features such as (flexible
variable selection rule), and theoretical guarantees for convex and
non-convex optimization. This is overall a good paper, which I
recommend to accept, even though its presentation could be a bit more
inviting and the experimental section could be improved.

Submitted by Assigned_Reviewer_35

This paper proposes a parallel block coordinate descent method. The objective functions considered here consist of convex terms which are separable and a smooth but possibly nonconvex term for which one uses a suitable approximation when minimizing coordinate-wise. As in [9] each coordinate-wise minimization is followed by an overrelaxation step. The contribution of this paper is to show that [9] can be used in conjunction with simple cyclic or randomized variable selection rules which allow to solve the subproblems in parallel.

This is definitely an interesting topic and can be applied in many machine learning applications. A weakness, however, is the experimental evaluation of the method. It remains unclear how the choice of the overrelaxation parameter effects the performance, e.g., if it makes sense to use diminishing step sizes. In Section 5, one should provide more details about the experimental setup to make the paper more self-contained instead of just citing [9]. For the sake of completeness, it should also be defined what the authors mean by "the classical serial BCD method". The runtimes for the larger Lasso problem are quite confusing. Parallelization seems to make almost no difference. This should be investigated in more detail, i.e., whether it depends on the hardware which is used or if there is an algorithmic way to exploit parallelism also in this case. In general, one could provide more numerical experiments with different minimization problems and different settings of the BCD method, e.g., comparing the different approximations mentioned in lines 118 and 119. A numerical comparison with [9] would also be interesting.
Furthermore, in line 58 it is mentioned that an inexact BCD method is proposed. It does not become clear in the theoretical results, however, in what sense inexact solutions of the subproblems are allowed.

Remark: The overrelaxation is also used in the paper "A Block Coordinate Descent Method for Regularized Multiconvex Optimization with Applications to Nonnegative Tensor Factorization and Completion" by Y. Xu and W. Yin.
Summary: Parallel BCD algorithms are an interesting topic and the paper seems mathematically sound. However, it is quite close to [9] and the numerical evaluation is not very convincing.

Submitted by Assigned_Reviewer_36

***** Update after author feedback ******
I agree with the authors' response and trust them to make the suggested changes that will improve the paper. On the other hand, as this added discussion won't be double checked by reviewers, I strongly encourage them to have the modification reviewed by a critical colleague in the field to ensure its quality.
*******************

quality: 7 (out of 10)
clarity: 8
originality: 6
significance: 7

SUMMARY: The authors propose a simple parallel scheme for the block-coordinate optimization of the sum of a smooth (possibly non-convex) function with a non-smooth convex block-separable function. The scheme finds a target point for each block in parallel in a chosen subset of blocks by minimizing the sum of a strongly convex approximation to the smooth part on this block (with matching gradients) and the non-smooth part. Each block in this subset is then updated (in parallel) as a convex combination of the previous value and the target points. A parallel proximal gradient scheme can be obtained as a special case; though using a convex combination of the iterates yield a slightly different scheme than previous work. The suggested algorithm is very similar to [9], except that in [9] the subset was chosen using a greedy scheme (which can be expensive), whereas this submission explores both randomized schemes or a cyclic scheme. For these, the authors prove the asymptotic convergence to a stationary point of the algorithm under standard Lipschitz gradient conditions. They also prove a O(1/k) convergence rate (where k is the number of cycles for the cyclic scheme) to the global minimum for the convex case or for the square of the norm of the objective proximal gradient for the non-convex case, for the algorithm with a constant step-size. They provide a succinct experimental comparison for a Lasso problem, observing that the cyclic scheme performs better than the randomized one, and that a speed-up can be obtained through parallelization (up to a point where the communication overhead starts to dominate).

EVALUATION:

I think this paper makes an interesting contribution to NIPS. There has been a lot of interest in parallel optimization schemes lately, and the one proposed in this paper, by analyzing general convex approximations to non-convex smooth objectives, is fairly general. Pieces of this submission has all been done somewhere else, but the combination is novel. For example, successive convex approximation schemes for non-convex problems was already analyzed in [24], but not for parallel updates. The contributions over [9] is to analyze the cyclic and randomized subset selection scheme, as well as to provide global convergence bounds (rather than just asymptotic convergence as in [9]). A contribution over several other parallel block coordinate schemes which was proposed recently is to consider the general successive convex approximation approach (instead of standard proximal gradient updates), which can give tighter bounds.

The paper is easy to follow, and the proofs are fairly self-contained. I was disappointed though that the authors didn't provide the proof for the cyclic updates for Lemma 3 (line 313-314) in supplementary material.

The numerical experiments are mainly a sanity check, and don't necessarily provide that much insights. The proof of asymptotic convergence (p. 5 mainly, as p.4 is used in other proofs) don't provide much insights and is fairly technical -- I would have suggested to put this part in an appendix in the supplementary material, and use the extra page freed to actually discuss more the meaning of their convergence rate results and how they compare with all the related schemes proposed in the literature. For example, how do the constants compare; what kind of speed-up is expected for different standard scenarios; what insights do the convergence analysis provide for practical usage of their algorithm (especially for a NIPS audience); etc. I think this would make the paper more useful to the NIPS community.

Pros:
- Simple and general parallel optimization scheme for smooth (possibly non-convex) objective with non-smooth separable part. The generality of the setup could be useful for the types of problems considered by the NIPS community.
- Fairly self-contained (and easy to read) convergence analysis for both convex and non-convex case, including both cyclic and randomized subset schemes.

Cons:
- No discussion providing insights about their convergence results or how they compare with other results in the literature.
- Toy empirical comparison.

== Other comments ==

Typos:
- Line 167, \tau_min is not defined.
- Line 253: the limit should be removed in this sentence, i.e. "... we have h(...) < = h( ...) for all x_i ..., *for all j*.
Summary: A nice complete convergence analysis for more practical subset selection schemes for the simple general parallel optimization algorithm proposed in [9]. More discussion / interpretation of the analysis in the paper might make it more useful to the NIPS community, though it is already fairly readable for the optimization specialists.
Author Feedback
Author rebuttal: General comments to all the reviewers and the area chair: Thanks a lot for your constructive comments. In the original manuscript, our goal was to keep all the paper information in the main body of the manuscript. However, based on the reviewers' comments, we have decided to move some parts of the paper to the supplementary materials to make more room available for some details on the convergence of the cyclic rule and the details of the numerical experiments. More specifically, we will move some parts of the proof of Theorem 1 and other proofs to the supplementary materials.
Regarding the numerical experiments in the paper, the reason that we choose LASSO as empirical example is that it has become a standard comparison example in machine learning community due to its simplicity and applications. A standard problem generation method is also used (instead of using real data) for the sake of simplicity and being easily reproducible. We have some extra set of numerical experiments and insights on the convergence of the algorithm (such as computation and communication overhead of the algorithm) that we will put in the supplementary materials to make the numerical experiments stronger. Although we agree with the reviewers that it is interesting to further investigate the numerical performance of the algorithm, due to space limitations (especially after resolving the formatting issue raised by one of the reviewers and adding some cyclic details), we will not be able to add more numerical experiments to the main body of the manuscript. Furthermore, we believe that the major contribution of the paper is on the theoretical results. As stated in the introduction section, the proposed algorithm is the first parallel method with iteration complexity guarantees which could handle non-smooth non-convex problems.

Below is our tailored response to the comments of each reviewer:

Response to the comments of referee "Assigned_Reviewer_19":
Thanks a lot for understanding our contributions. In the revised manuscript, as discussed above, we can easily shorten the length of the paper by moving some parts of Theorem 1 and other proofs to the supplementary materials and referring to other works for proofs if possible (as suggested by the reviewer). Therefore, we will be able to include the main steps of Lemma 3 for the cyclic rule in the main body and the complete details in the supplementary materials.

Response to the comments of referee "Assigned_Reviewer_35":
Thanks a lot for your careful review. Regarding the contributions of the work, in addition to what the reviewers mentioned, we believe that the results of this work are much more comprehensive than the existing work [9]. More precisely, this work not only studies the asymptotic behavior of the algorithm, but also considers more insightful analyses such as non-asymptotic and iteration complexity bounds of the algorithm. Therefore, we respectfully disagree with the reviewer's comment that "it is quite close to [9]". Our convergence results goes beyond [9] and our extensions are clearly non-trivial and not similar to [9].
Regarding the choice of the parameters, we have some extra numerical experiments that we will put in the supplementary materials. In short, there is a tradeoff for choosing the value of parameters \gamma and \tau. The smaller values of \gamma will result in less zigzag behavior for the convergence path of the algorithm; however, as a direct consequence, we have shorter steps which will slow down the convergence rate. In order to make the surrogate function sufficiently strongly convex, we need to choose \tau large enough; however, choosing too large tau values enforces the next iterates to stay close to the current iterate and results in slower convergence. Also it makes sense to consider diminishing step-size if the parameters of the objective function is not known a-priori.
Regarding your comment on the trade-off on increasing the number of processors, as mentioned in the paper, increasing the number of processing nodes, will increase the communication overhead in our machines. We have some additional numerical experiments which will be added to the supplementary materials showing how the communication overhead becomes a performance limiting factor in the algorithm in our machines.

Also on the numerical experiments details, as discussed above, we can obtain more space and therefore, we can explain the numerical experiments in more details. Furthermore, in our experiments, we have also tried to implement the method in [9]. However, there are different ways to implement the algorithm in [9] especially for the greedy selection search, the message passing method between the nodes, and the storage of the matrix A. Unfortunately, in our implementation we could not reproduce the numerical results in [9] and hence we think it is not reasonable to put the results in the paper -unless we have its package.

We call the algorithm "inexact" since it minimizes an approximation of the original objective function at each step instead of minimizing the exact original objective function.

Response to the comments of referee "Assigned_Reviewer_36":
Comments: Thanks a lot for your extremely comprehensive review which helped us a lot in improving the quality of the manuscript. We completely agree with your comment of having more insightful discussions and comparison with other results. We will do our best to include more discussions on the results which are more interesting to the NIPS community such as the parameter selection rules (see, e.g., response to the second reviewer) and the difference with other existing bounds. This, as mentioned earlier, will be achieved by making room in the main body of the manuscript by moving some parts to the supplementary materials.